# A single, continuous metric to define tiered serum neutralization potency against HIV

Peter Hraber[1]*, Bette Korber[1,2], Kshitij Wagh[1], David Montefiori[3], Mario Roederer[4]

[1]Theoretical Biology and Biophysics, Los Alamos National Laboratory, Los Alamos, United States; [2]New Mexico Consortium, Los Alamos, United States; [3]Department of Surgery, Duke University Medical Center, Durham, United States; [4]Vaccine Research Center, National Institute of Allergy and Infectious Diseases, National Institutes of Health, Bethesda, United States

**Abstract** HIV-1 Envelope (Env) variants are grouped into tiers by their neutralization-sensitivity phenotype. This helped to recognize that tier 1 neutralization responses can be elicited readily, but do not protect against new infections. Tier 3 viruses are the least sensitive to neutralization. Because most circulating viruses are tier 2, vaccines that elicit neutralization responses against them are needed. While tier classification is widely used for viruses, a way to rate serum or antibody neutralization responses in comparable terms is needed. Logistic regression of neutralization outcomes summarizes serum or antibody potency on a continuous, tier-like scale. It also tests significance of the neutralization score, to indicate cases where serum response does not depend on virus tiers. The method can standardize results from different virus panels, and could lead to high-throughput assays, which evaluate a single serum dilution, rather than a dilution series, for more efficient use of limited resources to screen samples from vaccinees.

DOI: https://doi.org/10.7554/eLife.31805.001

*For correspondence:
phraber@lanl.gov

Competing interests: The authors declare that no competing interests exist.

## Introduction

Comparing antibody neutralization activity of different sera against genetically and antigenically diverse viral strains requires standardization. ID50 (or ID80) values, the inhibitory dilutions at which 50% (or 80%) neutralization is attained, are determined for a panel of viruses, using the TZM-bl neutralization assay (*Sarzotti-Kelsoe et al., 2014*). Serum breadth and potency are two measures used to characterize neutralization responses across virus diversity. Breadth is the proportion of pseudoviruses with an ID50 score above the threshold of detection, and potency is the geometric mean ID50 (*Hraber et al., 2014*; *Rademeyer et al., 2016*). At least half of the variation in neutralization assay results from large panels can be explained by the averaged responses per serum, Env, and the entire panel, overall (*Hraber et al., 2014*). Serum breadth and potency therefore depend strongly on the Env panels used, which can vary markedly between studies.

Virus neutralization sensitivity to panels of sera from chronically infected individuals represents a continuum (*Seaman et al., 2010*). To characterize Envs in tiers involves partitioning large neutralization panels into three or four groups with similar sensitivity (*Rademeyer et al., 2016*; *Seaman et al., 2010*). Antibodies able to neutralize only tier 1 (most sensitive) viruses are readily elicited by HIV Env gp120 immunogens, but such tier1 responses are not protective; in human vaccine efficacy trials, such responses have been unable to confer protection against the viruses that continue to fuel the pandemic (*Gilbert et al., 2010*; *Montefiori et al., 2012*). Tier 2 viruses are more difficult to neutralize than tier 1, and represent the majority of viruses that are transmitted to establish new infections

(*Rademeyer et al., 2016*; *Seaman et al., 2010*). Tier 3 viruses are the most resistant to neutralization.

One difficulty with the tiered scheme for labeling viruses (i.e. tiers 1A, 1B, 2, and 3) is that it simplifies a continuous distribution into three or four categories (*Seaman et al., 2010*), despite wide variation within each category. Moreover, while the system categorizes viruses, it does not help compare serum neutralization potency. For example, a serum that neutralizes one tier 3 virus but only a few tier 2 viruses might subjectively be designated a 'tier 3 neutralizing serum,' while one which neutralizes no tier 3 viruses but many tier 2 viruses a 'tier 2 serum.' The latter serum is likely more potent (protective) in real-world scenarios despite being designated with a lower tier. A metric to rate sera for neutralization potency would be useful, for example to down-select vaccine candidates for further evaluation in clinical trials. Such a metric should be objective and continuous, rather than category-based. It should also provide biologically meaningful and interpretable values that are consistent with expectations of tiered viruses from terminology used by practitioners in this field.

Here, we describe an objective, quantitative metric for serum classification, and apply it to characterize serum neutralization activity against both large and smaller panels of pseudoviruses. It uses logistic regression to establish a numerical value for a given serum, based on its ability to neutralize viruses of different tiers. We describe a statistically motivated Neutralization Potency (NP) score, which represents serum neutralization tier on a continuous, rather than categorical, scale. That scale is designed to be intuitively meaningful to HIV researchers, such that sera with a low score (near 1) are able to neutralize only tier1 viruses, while sera with scores ranging from 2 to 3 reflect increasing capacity to neutralize tier 2 and 3 viruses. A continuum of NP values enables comparisons between sera. Rather than suggesting most sera can neutralize tier 2 viruses, NP values can distinguish between, say, 'tier 2.1' and 'tier 2.5' sera, the higher score indicating a better neutralization outcome. The potency comparisons are similar to comparing geometric mean neutralization titers, but instead are represented in tier-like terms.

Because this approach is based on the outcome of yes-or-no neutralization evaluations from a single dilution of serum, it can be used to evaluate large numbers of sera in novel high-throughput designs. The examples here use a threshold ID50 of 1:50, that is a binary assignment of whether or not a serum neutralizes 50% of a virus at a dilution of 1:50. This means the NP could be calculated from a single serum dilution, as opposed to a full eight-point titration series. The metric lends itself to high-throughput methods to compare neutralization potencies of many sera.

## Results

To define a metric that can compare neutralization potencies of different sera, we assigned a single neutralization index (NI) value to each HIV-1 envelope-pseudotyped virus (Materials and methods). The logarithm of the geometric mean ID50 against 205 sera was linearly rescaled to correspond with tier designations. The resulting NI values thus capture envelope neutralization sensitivity across 205 sera and provide a continuous scale of sensitivity that roughly corresponds to their tier designations. For logistic regression analyses below, the Envs were evaluated according to their NI. We note that the neutralization index could be defined in other ways, for example by using area-under-curve (AUC) values from the dilution series (*Yu et al., 2012*).

Tier-scaled virus NIs can be used to quantify serum neutralization activity. As higher-tier viruses are more difficult to neutralize, the expectation for a typical serum is that it can more potently neutralize lower tier viruses than higher tier viruses. In an over-simplification to illustrate the concept, we consider cases where a single threshold NI value cleanly separates outcomes, such that only those viruses above that NI resist neutralization by that serum. The serum would be assigned the threshold NI as a measure of neutralization potency. For example, one hypothetical serum that neutralizes tier 1 viruses up to tier 1.1 would receive a neutralization score of 1.1 (*Figure 1a*). Another hypothetical serum that neutralizes tier 2 viruses up to tier 2.8, receives a score of 2.8 (*Figure 1b*). Thus, in these simple examples, the scores for sera directly reflect their ability to neutralize viruses up to a rescaled virus tier value.

In practice, neutralization responses are noisy and not clearly resolved as in these two examples. Instead, the viruses neutralized by a particular serum are more scattered, and the overall trend becomes apparent in the aggregate view across many viruses, by considering how the probability of neutralization depends on rescaled tier values (*Figure 1c*). The probability for a serum to neutralize

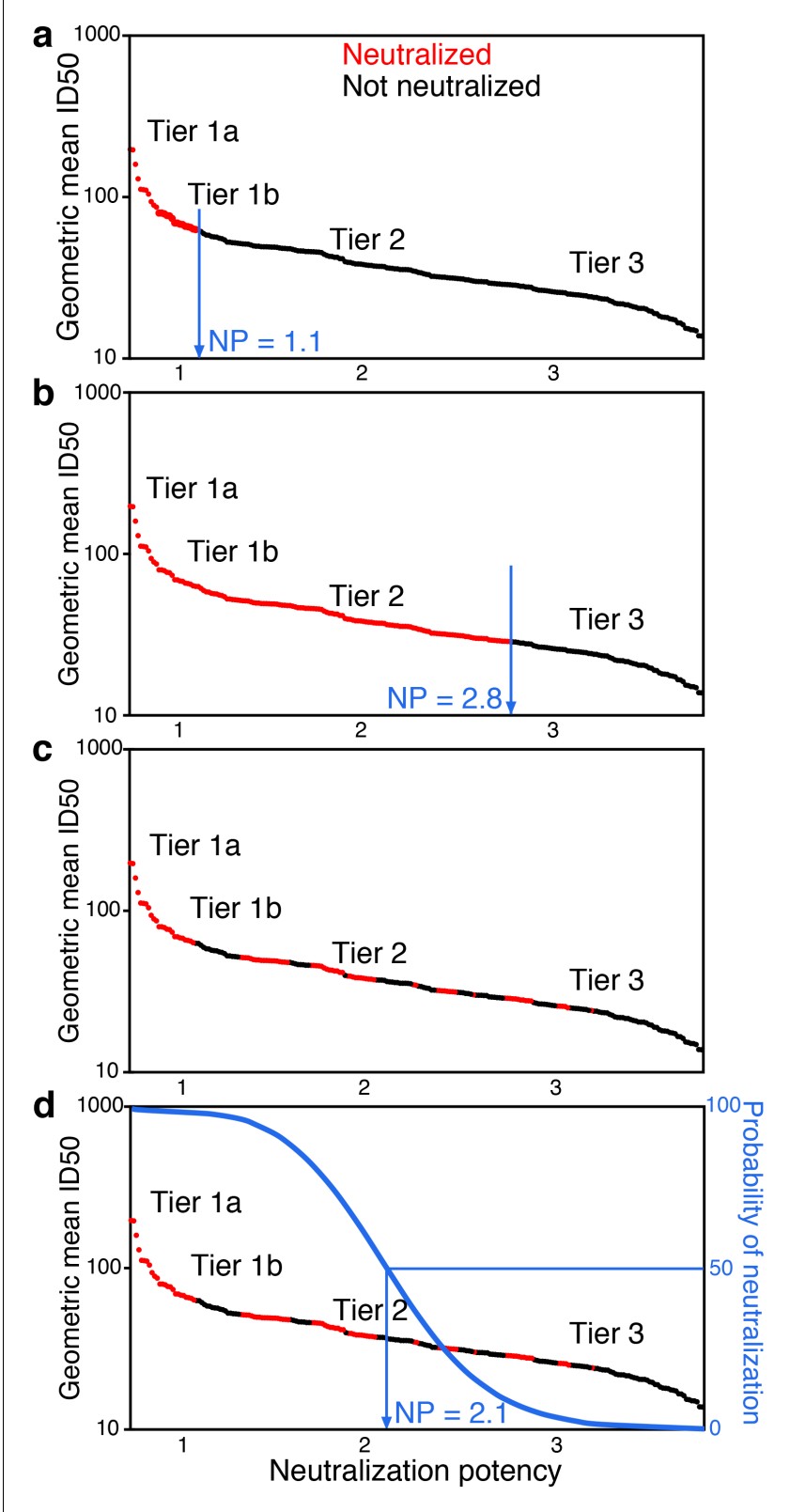

**Figure 1.** Conceptual introduction to serum neutralization potency (NP). (a) A hypothetical serum, which neutralizes tier 1A and some tier 1B viruses (red), but does not neutralize any tier 2 or 3 viruses (black), is assigned a neutralization potency (NP) of 1.1. (b) Another hypothetical serum may neutralize all tier 1A and B viruses and most tier 2 viruses, for NP of 2.8. In practice (c), the two outcomes do not segregate so clearly. Instead, positive

*Figure 1 continued*

and negative results among pseudoviruses are interspersed. Neutralization outcomes are scattered over the range of mean ID50s, and more sensitive viruses are enriched for positive neutralization. Logistic regression provides an objective way to distinguish neutralization outcomes. The neutralization outcome is treated as a probability (**d**). We use logistic regression to define the serum NP, which is the Env neutralization index (NI) value with 50% probability of neutralization that best separates neutralized and non-neutralized viruses.

DOI: https://doi.org/10.7554/eLife.31805.002

should decrease as virus NI values increase. We define serum neutralization potency (NP) as the NI that gives equal probability for viruses sensitive and resistant to neutralization by that serum, which we estimate using logistic regression (*Figure 1d* and Materials and methods).

## Serum neutralization potencies from 225 Envs

From the large, multi-clade, M group neutralization panel (*Hraber et al., 2014*), we computed log-geometric mean neutralization ID50 titers per virus and serum, then transformed the log-means onto the interval from at least 1 to below 4, to obtain tiered neutralization indices (NIs). This transformation gives an inverse relationship between tier-scaled NIs and geometric mean titer for viruses, as higher titers correspond to lower-tier viruses (*Figure 2a*).

Conversely for sera, higher titers averaged across viruses indicate greater probability of neutralizing higher-tier, neutralization-resistant viruses (*Figure 2b*). That is, tier 3 sera (NP >3) are better able to neutralize tier 3 viruses than are tier 1 sera (NP <2). A tier 2.5 serum should generally be able to neutralize viruses up to that point on the neutralization sensitivity continuum, although there could be some viruses that are neutralized above and a few resistant below. Low scatter, as illustrated in *Figure 1a and b*, gives a steep slope in logistic regression, indicating a sharp boundary between viruses neutralized and not neutralized. More scatter, that is lower contrast between viruses neutralized and not neutralized with increasing mean ID50, as illustrated in *Figure 1c*, gives a lower slope. Inability to resolve between neutralization outcomes would give no slope, quantified by a high probability of a false positive (p value) from rejecting the null hypothesis that the slope is zero, as is true for some NP outcomes in *Figure 2b*. This is most common at the low end of the serum neutralization continuum, where the ability to neutralize a virus is constant (and low) across the range of virus sensitivities.

Resampling (with replacement) sets of 225 Envs from the M group panel indicates that NP values are robust to sampling variation, save for a few sera with a slope of zero (*Figure 2—figure supplement 1a*). Variation may increase slightly among resampled NP values at the extremes of the neutralization scale even when the slope is non-zero (*Figure 2—figure supplement 1b*).

Neutralization responses for a typical serum, such as SA.C37, which has median potency among the sera we studied, appear in *Figure 3*. For each virus, the neutralization outcome is shown as a function of the tier-transformed geometric mean ID50 (*Figure 3a*). Serum neutralization potency computed from the 225 Env-panel is 2.5. Separation, with overlap, between viruses neutralized and not neutralized is apparent in *Figure 3b*.

## Serum neutralization potencies from small Env panels

As described in Materials and methods, we identified smaller panels and evaluated whether sets of 11 of the 12-Env global panel (*deCamp et al., 2014*), or either 10 or 20 Envs identified by lasso (*Tibshirani, 1996*; *Friedman et al., 2010*) could estimate neutralization indices more efficiently than the full set of 225 Envs. Together, five Envs were shared by all three small panels (*Figure 4—figure supplement 1*) and four additional Envs were common to both the global panel and 20-Env lasso panel, while the two hand-selected Envs were specific only to the global panel. The Envs that were selected to infer NP from smaller panels represent a range of neutralization sensitivities, favoring more sensitive and disfavoring the insensitive viruses (*Figure 4—figure supplement 1*).

A review of the ability of each panel to model NP for the serum SA.C37 (*Figure 4*) suggests that the 20-Env panel, as might be expected, is better able to resolve between neutralization outcomes (p=0.000162, *Figure 4c*) than smaller panels (*Figure 4a and b*; p=0.00298 and p=0.00151 for global

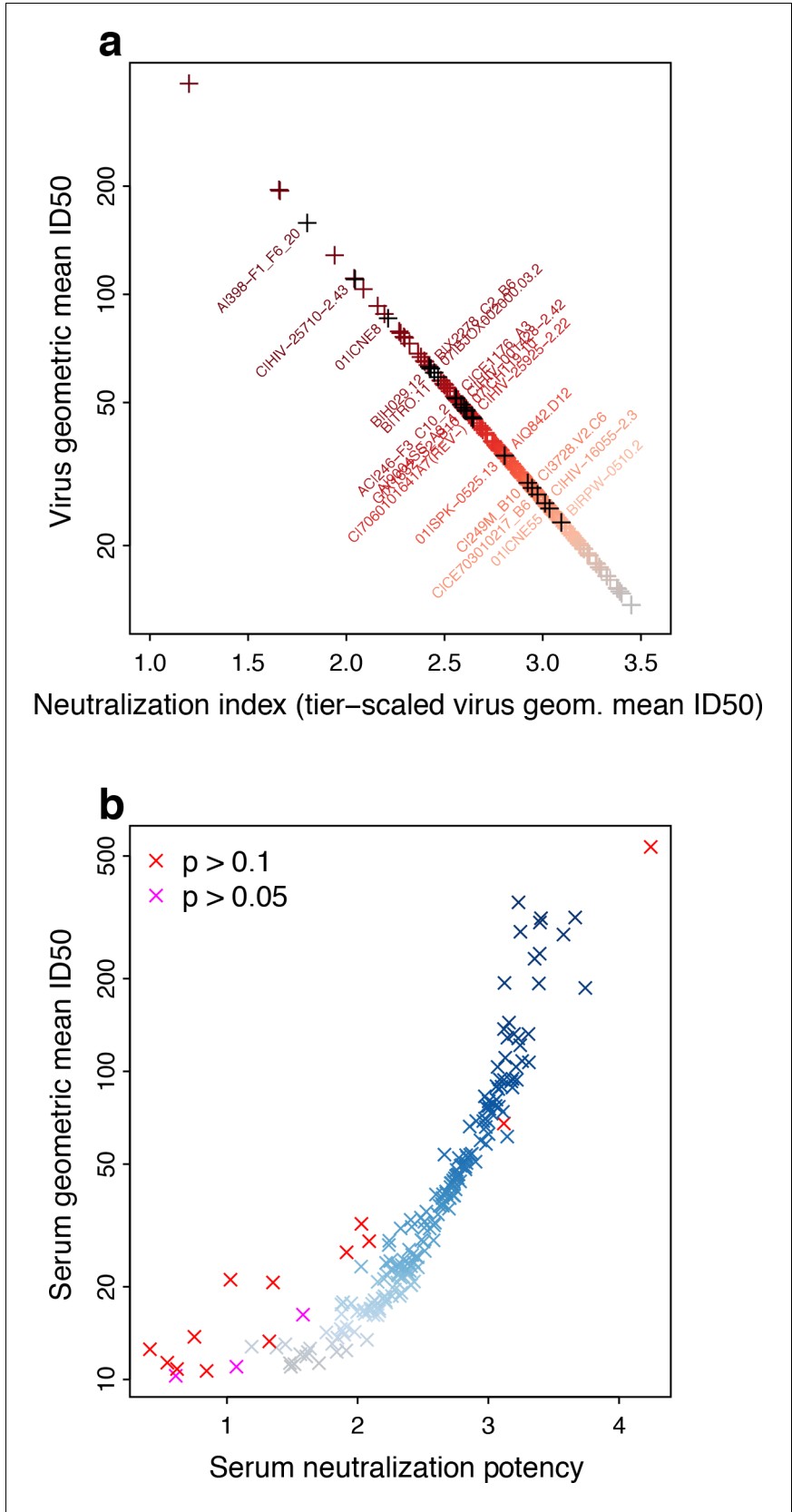

**Figure 2.** Neutralization potencies (NP) in the M-group panel of 225 Envs and 205 sera. (a) Linear transformation of NSDP virus geometric mean ID50 neutralization titers provides a tiered scale, based on previous reports. *Figure 2 continued on next page*

*Figure 2 continued*

Symbol colors indicate neutralization sensitivity, from ranked virus mean ID50s, and range from most (red) to least sensitive (grey). Envs identified for use in candidate subset panels that reproduce full virus panel NP values are labeled, with corresponding symbols colored black. We use the transformed values to compute serum NP. (**b**) Serum NPs are correlated with geometric mean ID50s per serum but, because of the transformation applied to viruses, range from about 1 to 4, consistent with the established Env tier classification scheme. Symbol colors show potency among ranked mean serum ID50s, and range from least (grey) to most potent (blue). Other colors indicate results from $\chi^2$ tests for non-zero slope, with Bonferroni corrections for 205 tests (red, experiment-wide p>0.1/205, that is per-comparison p>0.000488; magenta, experiment-wide p>0.05/205, per-comparison p>0.000244).

DOI: https://doi.org/10.7554/eLife.31805.003

The following figure supplement is available for figure 2:

**Figure supplement 1.** Comparison of observed and bootstrap-resampled neutralization potency values.

DOI: https://doi.org/10.7554/eLife.31805.004

and 10-Env panels, respectively). In this example, the standard global panel performed as well as the lasso panel of 10. This one case may not indicate the responses when tested across more sera.

NP values inferred from smaller panels are correlated with NP values from larger sets of holdout, non-panel Envs, and also with each other, across panels (*Figure 4—figure supplement 2*). The lasso-selected panel of 10 and the global panel perform similarly, so the global panel performs reasonably well for a panel of that size. The global panel may offer greater resolution for tier 1A and 1B sera, because it includes more Envs with NI below 2.0 than either of the panels selected by lasso. With panel Envs, inferred serum NP values were roughly limited to range from 1 to 4. When more Envs are tested, the NP values can fall below 1 or above 4, most likely because the most extreme Envs on the neutralization continuum allow the logistic regression to fit parameters outside the range typically expected.

Over all the 205 sera, the 20-Env panel is better powered to detect a non-zero slope than the smaller 10- and 11-Env panels (*Figure 4—figure supplement 3*), giving p>0.05 in about half as many cases as the smaller panels, likely because greater statistical power results from having nearly twice as many measurements to compute logistic regression parameters. Overall, rather than recommending use of a single panel to compute NP, it seems NP can be computed using a reasonable choice of Envs that represent a range of neutralization sensitivities, and use of more Envs is better able to quantify NP significantly than fewer Envs.

## Neutralization responses in progressors and long-term non-progressors

Using previously reported neutralization assay results (*Doria-Rose et al., 2010*), we computed geometric mean ID50 titers from 20 Envs and 103 donor sera, of which 25 were previously found to be long-term non-progressors (LTNP) (*Migueles et al., 2002*; *Migueles et al., 2008*; *Doria-Rose et al., 2009*). We identified 10 Envs in this panel that were also tested in the M group panel, two from subtype C (DU422.1, DU156.12), seven from subtype B (BG1168.1, 6101.10, TRJO4551.58, PVO.4, CAAN5342.A2, THRO4156.18, TRO.11), and one from subtype A (Q769.D22). We computed NP values from these 10 Envs, using a cutoff ID50 of 50 for positive neutralization outcome, and p-value cutoff from $\chi^2$ testing of 0.1 to indicate a significant NP score. This cutoff excluded 4 of 25 LTNP sera and 20 of 78 progressors; the proportions of excluded NP values were not significantly different between progressors and non-progressors (Fisher's exact test, p=0.42). NP values are highly correlated with geometric mean ID50s (*Figure 5*), regardless of whether or not NP outcomes with high p-values from $\chi^2$ testing are excluded (Kendall's τ, p<2.2 × 10$^{-16}$).

While the range of NP values in *Figure 5* may seem large, a score of 4.1 indicates sera that neutralized all ten Envs (ID50 ≥50), and the lowest-scoring sera neutralized none. To help interpret this range, consider the serum with a nominal NP of 4.6, which had a high corresponding p-value of 0.24. This particular serum neutralized 9 of 10 Envs, but the NP score is not significant. The only Env this serum failed to neutralize was DU156.12, which should be the most readily neutralized of these 10 Envs. (Its NI is 2.04, versus a mean NI of 2.91 and range from 2.46 to 3.34 among the other nine.) In this case, DU156.12 may contain one or more mutations that altered an epitope targeted by this serum.

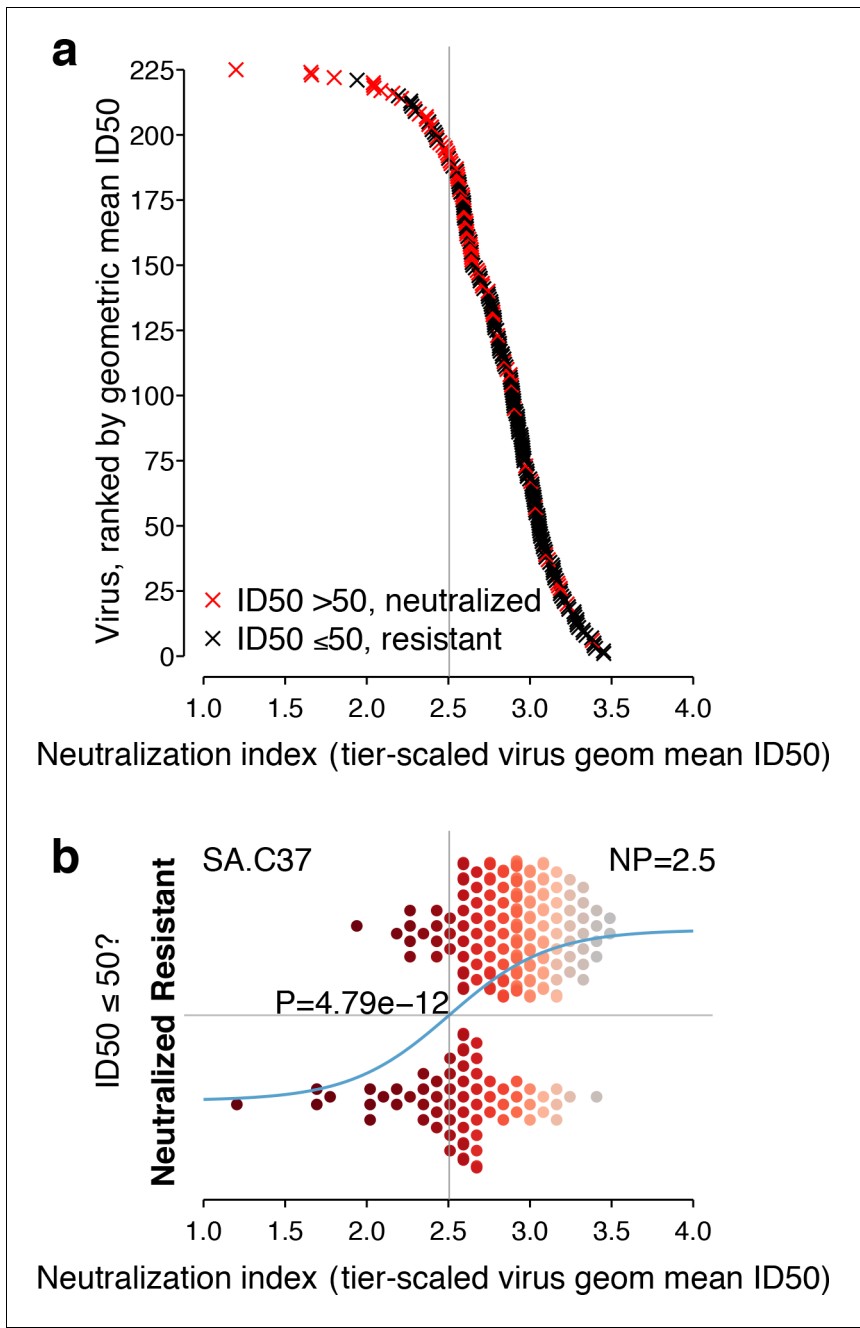

**Figure 3.** Neutralization outcomes and NP computation for a typical serum, SA.C37. This serum was chosen for illustration because it represents the median serum potency. (a) Outcomes for each of 225 viruses are either neutralized (ID50 >50, red) or not (ID50 ≤50, black) and are scattered noisily over virus mean ID50s, as in the hypothetical example (*Figure 1c*). (b) Beeswarm plot of the same data summarizes the NI distribution (tier-scaled geometric mean ID50 per virus) by outcome. The $\chi^2$ p-value for significance of the slope is $4.79 \times 10^{-12}$. A superimposed curve shows the inferred logistic function, and a vertical line indicates the NP at 2.5. Symbol color indicates virus neutralization sensitivity, as in *Figure 2a*.

DOI: https://doi.org/10.7554/eLife.31805.005

As anticipated, based on the established studies (*Doria-Rose et al., 2010*), the geometric mean ID50s differed significantly between the progressors and non-progressors (Wilcoxon test, p=$2.1 \times 10^{-11}$). We noted the same outcome for breadth, defined as the percentage of 20 Envs neutralized per serum with an ID50 of at least 50 (Wilcoxon p=$7.2 \times 10^{-11}$). Similarly, NP values

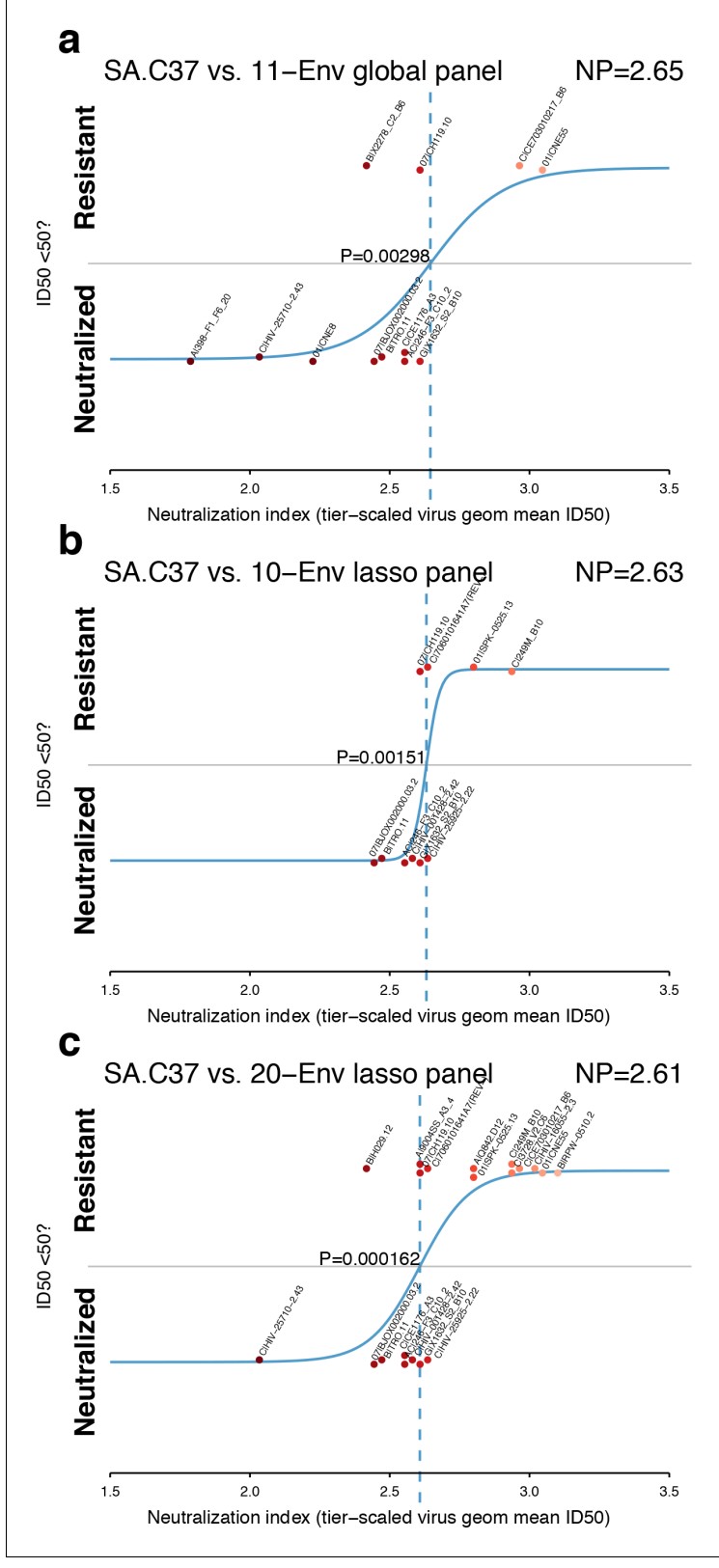

**Figure 4.** Panel-based NP estimates for a typical serum. The serum SA.C37 (*Figure 3*) was chosen for illustration because it represents median serum potency. (a) Global virus panel of 11 Envs. Lasso-selected (b) 10- and (c) 20- Env panels. In each case, panel viruses are identified by name, and text annotations indicate the NP (top-right corner), and the p-value for the null hypothesis of no slope (center).

*Figure 4 continued on next page*

*Figure 4 continued*

DOI: https://doi.org/10.7554/eLife.31805.006

The following figure supplements are available for figure 4:

**Figure supplement 1.** Heatmap of NSDP ID50 values identifies panel Envs.

DOI: https://doi.org/10.7554/eLife.31805.007

**Figure supplement 2.** Concordance of neutralization index estimates.

DOI: https://doi.org/10.7554/eLife.31805.008

**Figure supplement 3.** Cumulative p-value distributions for three candidate panels.

DOI: https://doi.org/10.7554/eLife.31805.009

---

differed significantly between these groups (n = 79, Wilcoxon p=4.3 $\times$ $10^{-9}$). This NP comparison therefore agrees with established findings (*Doria-Rose et al., 2010*), but notably, the NP comparisons involved half as many Envs and could use many fewer dilutions than required to compute mean ID50s among 20 Envs. To repeat the NP analysis using single-dilution assays, rather than a five-point (or greater) dilution series, the entire comparison would require at most one-tenth the number of neutralization reactions and material per sample, compared with the experiment using mean ID50 titers. Also, as seen above, testing more than 10 Envs per serum could increase statistical power, and reduce the number of sera excluded because their NP scores were deemed insufficiently significant.

## Vaccinee sera

Recent work to stabilize the clade A BG505 SOSIP.664 trimer and reduce conformational changes in the CD4-bound state has introduced mutations that increase hydrophobic packing of the V3 loop region, increase sensitivity to known neutralizing antibodies, and add disulfide bonds between gp120 and gp41 subunits within or between protomers. (*Sanders et al., 2013*; *Pugach et al., 2015*; *Julien et al., 2015*; *de Taeye et al., 2015*; *Ringe et al., 2017*) These mutations have been introduced to several Env isolates in addition to BG505, including a C-clade Env, ZM197M, to demonstrate general suitability of the approach (*Pugach et al., 2015*; *Julien et al., 2015*; *de Taeye et al., 2015*). A recent study (*Torrents de la Peña et al., 2017*) evaluated antigenicity and immunogenicity of these next-generation modified SOSIP trimers, designated v4 through v6.

We analyzed the post-vaccination serological data by computing neutralization potency scores from ID50 neutralization titers in 50 rabbits sampled 22 weeks after the first vaccination (boosted at weeks 4 and 20). Fifteen viruses from that study overlapped with the set for which we have already computed NI values, which we tabulate from most to least sensitive (*Table 1*). For comparison, we computed breadth as the fraction of these 15 Envs that were neutralized with an ID50 of at least 50 reciprocal dilutions. We computed geometric mean titers with censored values fixed at a low constant (i.e. <20 was treated as 10). Where two different laboratories tested the same Env, we used the more complete set of results (i.e. those with fewer missing values).

Results complement the original findings, and add to interpretation of the original assay results. Using a cutoff ID50 of 50, all but two of 50 rabbits tested yielded p-values below 0.05 (*Table 2*). These two animals (1586 and 1591 from Study C022-15, vaccinated with BG505 SOSIP.v5.1 and BG505 SOSIP.v5.2, respectively) neutralized only WITO, which is a relatively resistant Env. Comparing animals that showed similar breadth (7%, or 1 of 15 Envs neutralized) and potency (geometric mean ID50 of 13.3 and 12.8, respectively), the NP values from animals 1586 and 1591 are lower and of questionable significance (NP = 0.87, $\chi^2$ p=0.0831 for both) than animal 1588, vaccinated with BG505 SOSIP.v5.1 (NP = 1.73, $\chi^2$ p=0.0171).

Similarly, comparing outcomes in Study C0120-15 (*Table 2*), three of five animals vaccinated with ZM197M SOSIP.v5.2 (with IDs of 1875, 1876, and 1878) showed 20% breadth (3 of 15 Envs neutralized), geometric mean titers of 19.3, 19.1, and 21.9, and NP values above 2.0, with $\chi^2$ p-values below 0.05. This immunogen induced tier 2 responses in 4 of 5 rabbits, and yielded the most promising outcome among the refined SOSIP immunogens studied, though the clade-A BG505 trimers may have been at disadvantage because there were fewer clade-A Envs and A-related recombinants in the set utilized than clade-C Envs and C-related recombinants (*Table 1*). NP analysis agreed with the

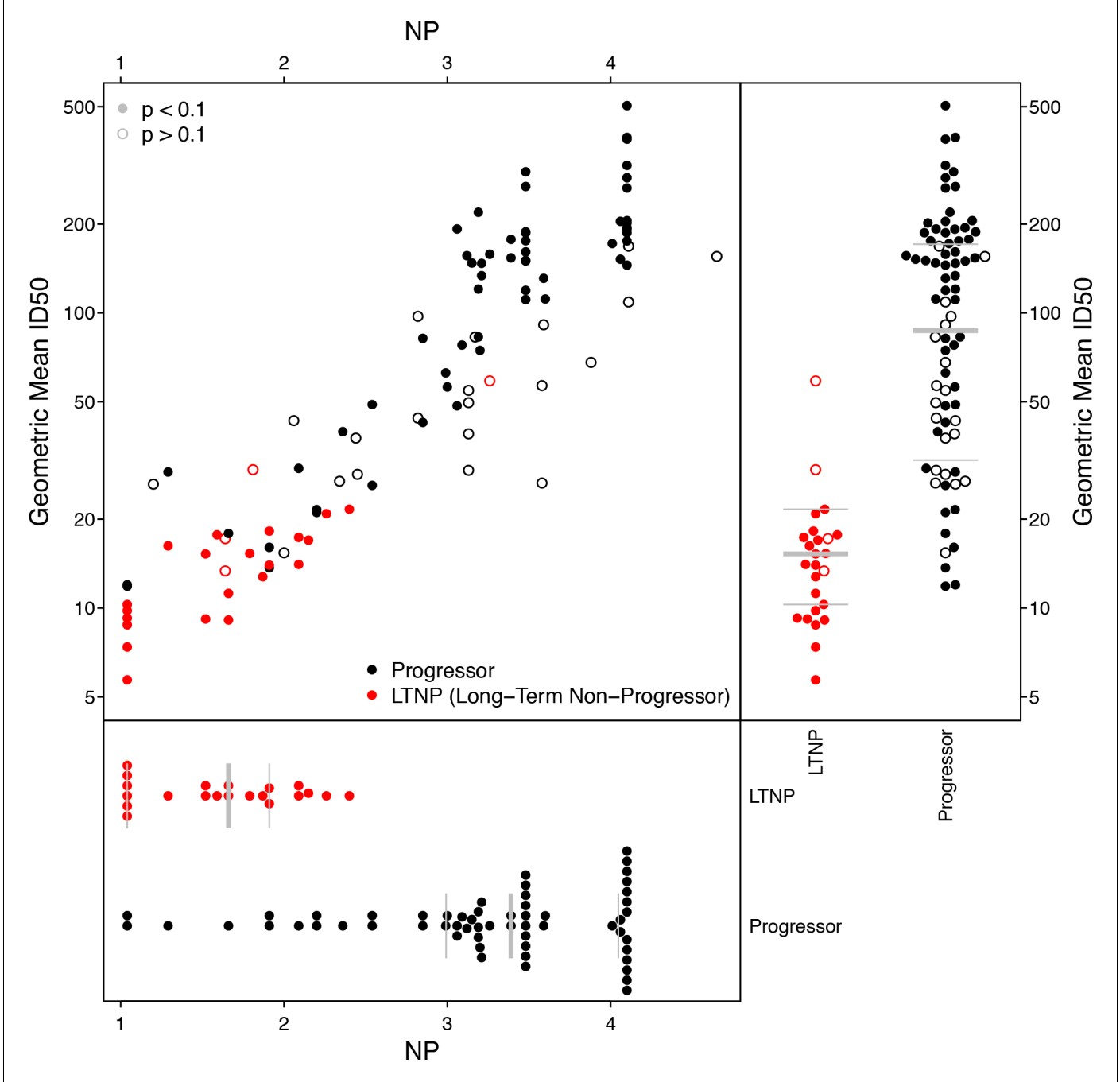

**Figure 5.** Neutralization potency analysis recapitulates mean titers and differences between progressors and long-term non-progressors (LTNP). Scatter-plot compares geometric mean ID50 titers, computed from 20 Envs, with NP scores, computed using 10 Envs. Symbol color shows whether the serum was from LTNP or progressor. Open circles had p-values from $\chi^2$ testing of 0.1 or more, suggesting the NP scores were unreliably quantified. Separate beeswarm plots show results for mean ID50 and NP scores, stratified by group.

DOI: https://doi.org/10.7554/eLife.31805.010

other neutralization metrics considered, and was able to resolve apparent ties between animals with similar neutralization responses to different Envs.

**Table 1.** Fifteen Env-pseudotyped viruses in neutralization assays against sera from 50 vaccinated rabbits sampled 22 weeks after initial vaccination (boosted at week 4 and 20) with stabilized SOSIP trimers, (*Torrents de la Peña et al., 2017*) utilized in *Table 2* to compare NP values using a cutoff ID50 of 50, breadth (% of Envs neutralized with an ID50 at least 50), and geometric mean titer (gmID50).

These data appear in Table S4 of the original paper (*Torrents de la Peña et al., 2017*). A dash in *Table 2* indicates ID50 below 20. Bold text in *Table 2* indicates positive neutralization outcomes in NP calculations.

| Column in *Table 2* | Name | Accession | NI | Subtype |
|---|---|---|---|---|
| a | 25710–2.43 | EF117271 | 2.04 | C |
| b | ZM197M | DQ388515 | 2.19 | C |
| c | ZM109F | AY424138 | 2.27 | C |
| d | TV1.21 | HM215437 | 2.32 | C |
| e | REJO | AY835449 | 2.36 | B |
| f | BJOX002000.03.2 | HM215364 | 2.45 | CRF07 |
| g | TRO.11 | AY835445 | 2.47 | B |
| h | CE1176_A3 | FJ444437 | 2.56 | C |
| i | 246-F3_C10_2 | HM215279 | 2.56 | AC |
| j | CH119.10 | EF117261 | 2.61 | CRF07 |
| k | X1632_S2_B10 | FJ817370 | 2.61 | B |
| l | ZM233M.PB6 | DQ388517 | 2.63 | C |
| m | WITO | AY835451 | 2.96 | B |
| n | CE703010217_B6 | KC894109 | 2.97 | A |
| o | CNE55 | HM215418 | 3.03 | CRF01 |

DOI: https://doi.org/10.7554/eLife.31805.011

## Antibody combinations

The same methods can be used with data from titration experiments using monoclonal antibodies. Although the scale of measurement is reversed (low IC50 values indicate high potency), a cutoff value can again be used to obtain a yes-or-no neutralization response. Using data from an earlier study (*Kong et al., 2015*), we explored the behavior of NP values from broadly neutralizing antibodies (bnAbs), both alone (monoclonal) and in combinations of up to four bnAbs. The NP and also the slope of the logistic function increase as bnAbs are combined in most cases, but not all (*Figure 6*). These results suggest that, all else equal, sera with higher number of antibody specificities could have higher NP and slope values.

## Discussion

We have described a simple method to quantify and compare serum neutralization probabilities. The method uses logistic regression to model the probability that a serum neutralizes a virus with an ID50 titer above some cutoff. The neutralization potency (NP) identifies where the probabilities of neutralizing and not neutralizing a virus are equal. It provides a continuous measure for sera, which builds upon established tier categories now used to rate virus sensitivity. The NP statistic defines the greatest virus tier that a serum can be expected to neutralize. Thus, an NP of (roughly) 1.8 to 2.8 is 'tier 2-like', and an NP above 2.8 is 'tier 3-like'. NP values below 1 are unable to neutralize even tier 1A Envs. The NP values are not absolute and depend on the ID50 cutoff used.

Defining a neutralization potency by testing a serum against a panel of 225 Envs on a routine basis is impractical and costly. We identified subsets of these Envs that largely reproduce the results from testing all 225. This makes assignment of NP values to a set of sera far more tractable than testing against all Envs. The already established 12-virus global panel may suffice to characterize NP values, although larger panels tend to give more significant outcomes.

**Table 2.** Comparison of NP, breadth, and geometric mean ID50 for Envs (*Torrents de la Peña et al., 2017*) listed in *Table 1*.

| Immunogen | ID | a | b | c | d | e | f | g | h | i | j | k | l | m | n | o | NP | P | Breadth | gmID50 |
|---|---|---|---|---|---|---|---|---|---|---|---|---|---|---|---|---|---|---|---|---|
| *Study C022-15* | | | | | | | | | | | | | | | | | | | | |
| BG505.664 | 1569 | - | - | - | 27 | 22 | - | - | - | - | - | - | - | - | - | - | 1.05 | 0.00582 | 0 | 11.3 |
| BG505.664 | 1570 | - | - | - | 28 | - | - | - | - | - | - | - | - | - | - | - | 1.05 | 0.00582 | 0 | 10.7 |
| BG505.664 | 1571 | - | - | - | **71** | - | - | - | - | - | - | - | - | - | - | - | 1.73 | 0.0171 | 7 | 11.4 |
| BG505.664 | 1572 | - | - | - | 30 | - | - | - | - | - | - | - | - | - | - | - | 1.05 | 0.00582 | 0 | 10.8 |
| BG505.664 | 1573 | - | - | - | **75** | - | - | - | - | - | - | - | - | - | - | - | 1.73 | 0.0171 | 7 | 11.4 |
| BG505.v4.1 | 1574 | - | - | - | **129** | 89 | - | - | - | - | - | - | - | - | - | - | 1.93 | 0.0212 | 13 | 13.7 |
| BG505.v4.1 | 1575 | - | - | 21 | - | - | - | - | - | - | - | - | - | - | - | - | 1.05 | 0.00582 | 0 | 10.5 |
| BG505.v4.1 | 1576 | - | - | - | - | - | - | - | - | - | - | - | - | - | - | - | 1.05 | 0.00582 | 0 | 10.0 |
| BG505.v4.1 | 1577 | - | - | 24 | - | - | - | - | - | - | - | - | - | - | - | - | 1.05 | 0.00582 | 0 | 10.6 |
| BG505.v4.1 | 1578 | - | - | 26 | 46 | 21 | - | - | - | - | - | - | - | 43 | - | - | 1.05 | 0.00582 | 0 | 13.7 |
| BG505.v5.1 | 1584 | - | - | 23 | 35 | - | - | - | - | - | - | - | - | 44 | - | - | 1.05 | 0.00582 | 0 | 12.7 |
| BG505.v5.1 | 1585 | - | - | 21 | - | - | - | - | - | - | - | - | - | - | - | - | 1.05 | 0.00582 | 0 | 10.5 |
| BG505.v5.1 | 1586 | - | - | 24 | 27 | 20 | - | - | - | - | - | - | - | **58** | - | - | 0.87 | 0.0831 | 7 | 13.3 |
| BG505.v5.1 | 1587 | - | - | - | - | 20 | - | - | - | - | - | - | - | - | - | - | 1.05 | 0.00582 | 0 | 10.5 |
| BG505.v5.1 | 1588 | - | - | 28 | **55** | 24 | - | - | - | - | - | - | - | - | - | - | 1.73 | 0.0171 | 7 | 12.7 |
| BG505.v5.2 | 1589 | - | - | 28 | 30 | - | - | - | - | - | - | - | - | - | - | - | 1.05 | 0.00582 | 0 | 11.5 |
| BG505.v5.2 | 1590 | - | - | 24 | 22 | 27 | - | - | - | - | - | - | - | 23 | - | - | 1.05 | 0.00582 | 0 | 12.6 |
| BG505.v5.2 | 1591 | - | - | 22 | 31 | - | - | - | - | - | - | - | - | **60** | - | - | 0.87 | 0.0831 | 7 | 12.8 |
| BG505.v5.2 | 1592 | - | - | 23 | 28 | - | - | - | - | - | - | - | - | 21 | - | - | 1.05 | 0.00582 | 0 | 11.9 |
| BG505.v5.2 | 1593 | - | - | 24 | 24 | - | - | - | - | 33 | - | - | - | - | - | - | 1.05 | 0.00582 | 0 | 12.2 |
| *Study C0119-15* | | | | | | | | | | | | | | | | | | | | |
| BG505.v5.2 | 1819 | - | - | - | - | - | - | - | - | - | - | - | - | - | - | - | 1.05 | 0.00582 | 0 | 10.0 |
| BG505.v5.2 | 1820 | - | - | - | - | - | 21 | - | 21 | - | 26 | 23 | - | - | 25 | - | 1.05 | 0.00582 | 0 | 13.2 |
| BG505.v5.2 | 1821 | - | - | - | 30 | - | - | - | - | - | - | - | - | - | - | - | 1.05 | 0.00582 | 0 | 10.8 |
| BG505.v5.2 | 1822 | - | - | - | - | - | - | - | 25 | - | - | - | - | - | - | - | 1.05 | 0.00582 | 0 | 10.6 |
| BG505.v5.2 | 1823 | - | - | - | 35 | - | 26 | - | - | 28 | 36 | 25 | - | - | 30 | 24 | 1.05 | 0.00582 | 0 | 16.4 |
| v5.2+211 C-433C | 1824 | - | - | - | **72** | - | - | - | - | - | - | - | - | - | - | - | 1.73 | 0.0171 | 7 | 11.4 |
| v5.2+211 C-433C | 1825 | - | - | **76** | 30 | - | - | - | - | - | - | - | - | - | - | - | 1.88 | 0.0131 | 7 | 12.3 |
| v5.2+211 C-433C | 1826 | - | - | 46 | 30 | - | - | - | 22 | - | - | - | - | - | - | - | 1.05 | 0.00582 | 0 | 12.6 |
| v5.2+211 C-433C | 1827 | - | - | **75** | 50 | 66 | - | - | - | - | 27 | - | - | 40 | - | - | 2.00 | 0.0167 | 20 | 16.9 |
| v5.2+211 C-433C | 1828 | - | - | 37 | - | - | - | - | - | - | - | - | - | - | - | - | 1.05 | 0.00582 | 0 | 10.9 |
| BG505.v6 | 1829 | 29 | - | 36 | 42 | - | 25 | - | 24 | 30 | 27 | 25 | - | - | 26 | - | 1.05 | 0.00582 | 0 | 18.9 |
| BG505.v6 | 1830 | 31 | - | **80** | **97** | 42 | 29 | - | 26 | 32 | 29 | 31 | - | - | 29 | 21 | 2.04 | 0.0134 | 13 | 25.6 |
| BG505.v6 | 1831 | - | - | **160** | **162** | 22 | - | - | - | - | - | - | - | - | - | - | 2.04 | 0.0134 | 13 | 15.3 |
| BG505.v6 | 1832 | 29 | - | **65** | **468** | 47 | - | - | - | - | - | - | - | - | - | - | 2.04 | 0.0134 | 13 | 17.4 |
| BG505.v6 | 1833 | - | - | 47 | 23 | 33 | - | - | - | - | - | - | - | - | - | - | 1.05 | 0.00582 | 0 | 12.7 |
| *Study C0045-15* | | | | | | | | | | | | | | | | | | | | |
| ZM197M.664 | 1649 | 26 | **162** | 23 | 35 | - | - | - | 32 | 37 | 37 | - | - | - | 22 | | 2.01 | 0.00765 | 7 | 20.0 |
| ZM197M.664 | 1650 | - | - | - | 20 | - | - | - | - | - | - | - | - | - | - | - | 1.05 | 0.00582 | 0 | 10.5 |
| ZM197M.664 | 1651 | - | - | - | 35 | - | - | - | 21 | - | - | - | - | - | - | - | 1.05 | 0.00582 | 0 | 11.4 |
| ZM197M.664 | 1652 | 23 | **365** | - | 35 | - | - | - | 45 | 24 | 27 | - | - | - | - | - | 2.01 | 0.00765 | 7 | 18.3 |
| ZM197M.664 | 1653 | - | 21 | 31 | 38 | - | - | - | - | - | - | - | - | - | - | - | 1.05 | 0.00582 | 0 | 12.4 |
| ZM197M.v4.2 | 1654 | - | - | - | - | - | - | - | 21 | - | - | - | - | - | - | - | 1.05 | 0.00582 | 0 | 10.5 |

*Table 2 continued on next page*

*Table 2 continued*

| Immunogen | ID | Env | | | | | | | | | | | | | | | NP | P | Breadth | gmID50 |
|---|---|---|---|---|---|---|---|---|---|---|---|---|---|---|---|---|---|---|---|---|
| ZM197M.v4.2 | 1655 | - | **4860** | - | 21 | - | - | - | - | - | - | - | - | - | - | - | 2.01 | 0.00765 | 7 | 15.9 |
| ZM197M.v4.2 | 1656 | 25 | 28 | 20 | **69** | - | - | - | - | - | - | - | - | - | - | - | 1.73 | 0.0171 | 7 | 13.6 |
| ZM197M.v4.2 | 1657 | - | - | - | - | - | - | - | - | - | - | - | - | - | - | - | 1.05 | 0.00582 | 0 | 10.0 |
| ZM197M.v4.2 | 1658 | - | - | - | 32 | - | - | - | - | - | - | - | - | - | - | - | 1.05 | 0.00582 | 0 | 10.8 |
| *Study C0120-15* | | | | | | | | | | | | | | | | | | | | |
| ZM197M.v5.2 | 1874 | - | **1010** | 49 | **53** | - | 21 | - | - | 27 | - | - | - | - | - | - | 2.11 | 0.00828 | 13 | 19.0 |
| ZM197M.v5.2 | 1875 | - | **153** | 87 | 60 | 45 | 23 | - | - | 24 | - | - | - | - | - | - | 2.23 | 0.00389 | 20 | 19.3 |
| ZM197M.v5.2 | 1876 | 24 | **64** | 35 | **69** | 68 | 20 | - | - | 33 | - | - | - | - | - | - | 2.19 | 0.0079 | 20 | 19.1 |
| ZM197M.v5.2 | 1877 | - | 45 | 20 | 37 | - | 25 | - | - | 38 | 22 | 25 | - | - | 27 | 25 | 1.05 | 0.00582 | 0 | 18.7 |
| ZM197M.v5.2 | 1878 | - | **114** | 21 | **68** | - | 31 | - | - | **52** | 29 | 25 | - | - | 27 | 25 | 2.07 | 0.0219 | 20 | 21.9 |

DOI: https://doi.org/10.7554/eLife.31805.012

Evaluating neutralization assay outcomes against the continuum of neutralization sensitivity among viral variants provides more context to interpret results, because it considers not only the proportion of tier 2 Envs neutralized (breadth), but which Envs should most likely be neutralized. This helps to interpret differences between sera that neutralize the same number of Envs, each of which have different sensitivities. It also helps to compare sera where titers may be averaged over many outcomes below the limit of assay quantification, as was the case for sera from vaccinated rabbits (*Table 2*) (*Torrents de la Peña et al., 2017*).

We also explored results from experiments that utilized different Env panels and found they can be compared on the same neutralization scale (not shown). The ability to do this requires only that some number of Envs in each panel have available tiered neutralization scores, computed from available data. Better comparative results are obtained when using more Envs, because resulting NP values are less likely to be undefined.

A web-based utility – http://hiv.lanl.gov/content/sequence/NI/ni.html – at the Los Alamos HIV database computes NPs for sera tested against subsets of M group Envs. In addition to the analysis described here, it can also compute and report NPs for clade C Envs (*Rademeyer et al., 2016*) and for antibody IC50s.

Broadly neutralizing monoclonal antibodies are similarly characterized when isolated, and TZM-bl assay inhibitory concentration IC50 and IC80 scores are generally determined across large pseudovirus panels, for example, to characterize a newly isolated antibody (*Wu et al., 2010*). We applied the same analytic methods to IC50s from antibodies. Our analysis of data from experiments that combined broadly neutralizing antibodies (bnAbs) having distinct specificities suggests that the NP increases with the number or variety of distinct antibody specificities in the sample.

The NP, as defined here, is a single metric to compare serum potencies. However, logistic regression provides another parameter, the slope. The slope indicates agreement between serum neutralization outcomes and average potency (geometric mean ID50) among serum samples used to compute the NI per Env. We hypothesize that the slope should be low for sera with limited epitope breadth. In the extreme case of a serum that targets a single epitope (e.g. a monoclonal response dominates), only Envs with that epitope would be neutralized, and neutralization outcomes should be widely scattered among viruses tested, independent of an overall sensitivity of the virus to neutralization, resulting in a slope near zero. Thus, the slope may indicate diversity of epitopes targeted by the test serum. Consistent with this, we found single monoclonal bnAbs had lower slopes than mixtures of monoclonal bnAbs. Such a finding might help characterize the mixtures of antibody specificities in polyclonal sera, to complement the methods for computational neutralization fingerprinting that have recently been advanced (*Georgiev et al., 2013*; *Doria-Rose et al., 2017*). To evaluate this idea, subsequent work could identify serum samples with similar NP values but different slopes and map the epitopes therein.

In addition to establishing a metric for serum neutralization, a primary advantage of this approach is that it suggests a strategy to simplify neutralization assay experiments, for more cost-effective

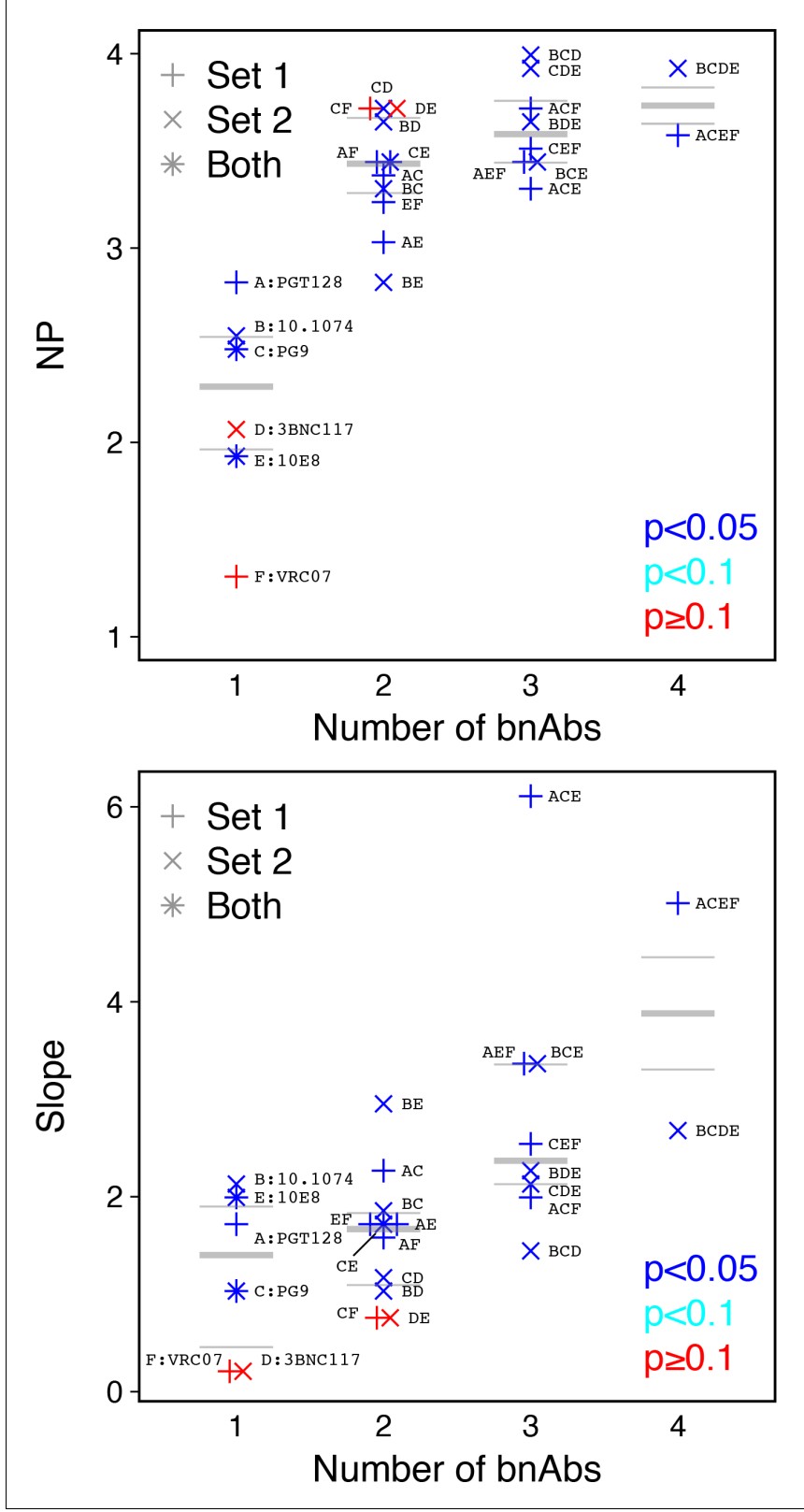

**Figure 6.** Analysis of monoclonal bnAb combinations. Increasing the number of bnAbs increases NP and slope. We used a cutoff IC50 of 0.1 μg/ml for 112 Envs and 27 bnAb combinations (**Kong et al., 2015**). (a) Neutralization potency (NP = $-b_0/b_1$, where $b_0$ is intercept and $b_1$ is slope of logistic function). (b) Slope ($b_1$) of logistic function. Up to four bnAbs were combined per set. Set 1 included PGT128, PG9, 10E8, and VRC07. Set 2 included 10.1074,

*Figure 6 continued on next page*

*Figure 6 continued*

PG9, 3BNC117, and 10E8. (PG9 and 10E8 were in both.) Letters A through F correspond to individual bnAbs and are used to label combinations, for example the four bnAbs combined in Set 1 are indicated as ACEF and in Set 2 as BCDE. p-Values indicate slope significance by $\chi^2$ test.

DOI: https://doi.org/10.7554/eLife.31805.013

screening of responses in large-scale vaccination studies. Logistic regression utilizes a collection of binary ('true' or 'false') responses, rather than the actual neutralization titers. Because its derivation relies only on a single-dilution analysis, rather than the current eight-point dilution series, it can enable a larger-scale screening throughput. Sera that score well using this screening metric could be prioritized for a more thorough dilution-series analysis. Thus, our approach provides a relatively simple metric by which serum neutralization of HIV viruses can be compared with a quantitative method. Together with the slope (which may indicate breadth of epitopes targeted), this approach could be useful to down-select vaccine candidates and move forward with regimens that are able to elicit, on average, greater NP values.

In summary, we propose a way to simplify the comparison of neutralization potency of antisera. The resulting metric is continuous, scaled to provide an easily interpreted value, and may provide a formal method for ranking antisera. It is used on single dilution assay data, lending it to a high-throughput platform.

## Materials and methods

### Env tier-scaled neutralization index (NI)

To obtain tier-scaled neutralization scores for sera, we first transformed for each virus the geometric mean ID50s against a panel of chronic sera to a range of values that correspond to tiers, based on previously published results from testing multiple sera against many different Envs (*Hraber et al., 2014*; *Rademeyer et al., 2016*; *Seaman et al., 2010*). In an earlier study that inferred tiers using Envs and unpooled sera (*Rademeyer et al., 2016*), the greatest geometric mean ID50 against tier 3 Envs was 26.7 and the greatest geometric mean ID50 against tier 2 Envs was 117.6. We used these two values to set the boundaries between tiers 2 and 3 and between tiers 1 and 2, respectively. A linear transformation then scaled the logarithm of the virus geometric mean ID50 to virus neutralization tier. (The log-mean is appropriate because the distribution of means is skewed, and log-transformation provides a more symmetric, normalized outcome.) We call this single transformed value for each virus the neutralization index (NI), and computed NI from Env geometric mean ID50s as 2 – [log(geometric mean ID50) – log(117.6)] / [log(117.6) – log(26.7)].

The NI is a continuously valued quantity, which can be interpreted intuitively on the tier scale, for example tier 1 viruses have higher mean neutralization ID50s and lower NI than tier 3 viruses. The resulting scale is not intended as an absolute or rigid standard, but rather as a guideline for interpretation. Such a transformation should best be held constant to compute and compare neutralization indices. The cutoffs would likely be different for different background neutralization data, such as a large Env panel against pooled sera (*Seaman et al., 2010*). Any linear transformation of the log-transformed geometric mean ID50s, or leaving them as they stand, would yield results identical to our findings, but on a modified numerical scale. The NI transformation makes it possible to interpret neutralization scores in terms of the familiar tier system.

### Serum neutralization potency (NP)

To define binomial (yes/no) outcomes for any serum, we set an ID50 threshold value, and consider a virus as neutralized or resistant to that serum, depending on whether or not ID50 was above the threshold, respectively. Here, we use a cutoff dilution of 1:50, though this could be changed as conditions merit. While 1:50 is a good working choice for sera from natural infection cases, in a vaccine setting, a more generous choice (say 1:20) might be desirable. Changing the cutoff could introduce inconsistent interpretations among results from different cutoffs. The M group neutralization data (*Hraber et al., 2014*) have a median ID50 of 28, and 37% of observations are above the 1:50 cutoff value.

For each serum, we use logistic regression to model the probability of neutralization as a function of the tier-transformed virus neutralization score (NI), using the glm function in R (version 3.4.0). In cases where parameter estimation did not converge on a solution, we used the function glm2 (version 1.1.2) and more iterations, to ensure optimization converges on the solution. The glm2 function was designed to overcome the convergence limitations of its predecessor (*Marschner, 2011*).

The logistic function is defined as $p(x)=1/[1 + \exp(b_0 + b_1 x)]$, with $x$ the independent variable, and two parameters: intercept ($b_0$) and slope ($b_1$). We define neutralization potency (NP) as $-b_0/b_1$. This corresponds to the NI that has equal odds of being neutralized or not (*Crawley, 2002*). Thus, the NP assigns each serum a tier-transformed neutralization score, which best separates the neutralized and non-neutralized viruses. In practice, we found it useful to include two hypothetical viruses with extreme phenotypes, one always sensitive to neutralization by any serum, with a tiered score of 0, and one always resistant to neutralization, with a tiered score of 5. These extremes help to define the NP and ensure the regression calculations perform as expected.

An important caveat remains to be addressed. Because it would require division by zero, NP is undefined if the slope is zero. This occurs when the probability of neutralization is independent of the tier-transformed NI values for viruses (i.e. is a constant equal to the breadth of the serum), meaning there is no consistent NI value that can separate neutralized from non-neutralized viruses. This might result in cases of low statistical power, a non-representative selection of viruses (they are assumed random and independent), or a serum with a response that is otherwise atypical of sera used to compute mean virus neutralization titers. In such cases, one might report the slope and intercept separately, and refrain from interpreting NP. Because the slope is a statistical inference, a formalism exists to evaluate the null hypothesis of no slope. This is achieved by a likelihood-ratio test, which computes a p-value from the $\chi^2$ distribution for the reduction in deviance that results from adding a slope to the regression model (*Crawley, 2002*). A small p-value indicates a significant, non-zero slope, and that the NP is well defined. NP values with high p-values should be interpreted with caution.

## Panel selection and validation

It would be impractical to require ID50s from 200 distinct Envs to compute serum NP values. We sought to develop and validate smaller, representative Env panels useful to estimate serum NP. Recently a global panel of 12 viruses was developed for its ability to model the median of the distribution of the magnitude-breadth curve (*deCamp et al., 2014*). That panel was selected using lasso to identify nine Envs for their ability to model the median area under the dilution-series curve (AUC) for many sera. To these Envs, three were added manually, to include some neutralization response profiles not included in the nine. Here, we used this Env set as a candidate panel to infer the full-panel NPs. Because of missing values among the NSDP panel data, we omitted one virus (clade A, 398-F1_F6_20), which was missing values from 39 of 205 sera. We did this because the missing value would have caused different effective sample sizes across sera and may have reduced the apparent robustness of resulting NP values. This procedure yielded a panel of 11 Envs.

Because our approach to compute NP values uses mean ID50 as an input variable, we again used lasso (*Tibshirani, 1996*) to select alternative small panels. Here, instead of modeling median AUC, we sought predictors to model the logarithm of the geometric mean ID50 per serum, using the glmnet R package(*Friedman et al., 2010*), version 2.0–10, to obtain panels of 10 and 20 Envs. We used bootstrap resampling (with replacement) to assess NP robustness, and summarized the results as median and inter-quartile range per serum. To evaluate panel robustness, we compare the NP values from the panel and the remaining held-out Envs, testing for correlations between them. We also evaluated correlations among NP values from alternative panels, and the distribution of p-values from logistic regression.

## Acknowledgements

We thank Gabriella Scarlatti, affiliates of the Global HIV Vaccine Enterprise, and participants of the workshop on appropriate use of tiered virus panels when assessing HIV-1 vaccine-elicited neutralizing antibodies, for the discussions that inspired this study. Nicole Doria-Rose and Mark Connors kindly shared their serum neutralization data. We thank authors of the other cited reports for making their neutralization data available at the time of publication. This work was supported by the Bill and

Melinda Gates Foundation [OPP1146996]. MR was supported by the Intramural Research Program of the Vaccine Research Center (VRC), National Institute of Allergy and Infectious Disease (NIAID), National Institutes of Health (NIH).

## Additional information

### Funding

| Funder | Grant reference number | Author |
|---|---|---|
| Bill and Melinda Gates Foundation | OPP1146996 | Peter Hraber<br>Bette Korber<br>Kshitij Wagh<br>David Montefiori |
| National Institute of Allergy and Infectious Diseases | | Mario Roederer |

The funders had no role in study design, data collection and interpretation, or the decision to submit the work for publication.

### Author contributions

Peter Hraber, Conceptualization, Software, Formal analysis, Validation, Investigation, Visualization, Methodology, Writing—original draft, Writing—review and editing; Bette Korber, Conceptualization, Resources, Supervision, Funding acquisition, Validation, Investigation, Methodology, Writing—original draft, Writing—review and editing; Kshitij Wagh, Conceptualization, Formal analysis, Validation, Investigation, Methodology, Writing—review and editing; David Montefiori, Conceptualization, Resources, Supervision, Funding acquisition, Investigation, Project administration, Writing—review and editing; Mario Roederer, Conceptualization, Resources, Software, Formal analysis, Validation, Investigation, Visualization, Methodology, Writing—original draft, Writing—review and editing

### Author ORCIDs

Peter Hraber http://orcid.org/0000-0002-2920-4897

### Decision letter and Author response

Decision letter https://doi.org/10.7554/eLife.31805.016
Author response https://doi.org/10.7554/eLife.31805.017

## Additional files

### Supplementary files

• Transparent reporting form
DOI: https://doi.org/10.7554/eLife.31805.014

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
