## [Decision Letter]

Congratulations, we are pleased to inform you that your article, "A Single, Continuous Metric to Define Tiered Serum Neutralization Potency against HIV", has been accepted for publication in *eLife*.

This well written manuscript from Hraber and colleagues describes a novel method for applying a single score to quantify polyclonal antibody responses directed to HIV Envelope that result in blocking of virus in vivo, called virus neutralization. This approach of classifying HIV-positive serum samples is analogous to what has been done for viral isolates. Neutralization results from the combined activities of multiple single monoclonals in a given individual, each of which possesses potentially different affinities and activities. It has been a challenge for the vaccine field to evaluate the activity of antibodies in a consistent manner. The proposed approach will be extremely useful in providing an objective, quantitative, high throughput approach for evaluating the potential of vaccine regimens geared towards neutralizing antibodies and/or classifying patient plasma or antibody potency and breadth. Advantages of this approach include: factoring in the sensitivity of the variants; cutting down on the time and cost needed to perform these evaluations; and importantly enhance standardized comparisons between studies.

The statistical approach consists of cascading the neutralization sensitivity level of HIV-1 variants to generate a neutralization index score (NI) to report a serum neutralization potency (NP) score from 1 to 4, with 1 being the most sensitive variants and scores > 2 being able to neutralize the more relevant patient derived variants. In summary, a vaccine approach that achieves a higher potency score would be considered more promising and worthy of further investigation, compared to those that elicited lower scores.

The authors provide a link to a tool that they will make available for calculating NP on the Los Alamos HIV Database website. This tool includes pre-calculated NI values for several hundred envelope variants that are widely available and would be commonly used by labs that do this type of work. The acceptability of IC50 or binary data increases the utility of the tool.

The value of this approach cannot be underscored sufficiently, as it has been extraordinarily challenging to compare serum responses from infected humans, monkeys and other animals immunized with various vaccines or infected with SHIV (monkeys) or HIV (humans). At present, the evaluation of neutralizing antibodies requires that a set of virus isolates be tested individually, as the authors note, in a dilution series, in a cumbersome although painstakingly validated assay. Despite the use of common 'panels' of virus it is difficult to interpret hence the need for a simpler method to advance the field which these authors do. The proposed approach also works on smaller virus panels and may provide a higher-throughput and more cost-effective method for screening sera. Importantly, as part of the development of the methodology, they have studied the longitudinal responses in HIV-infected subjects (n=2 shown in the manuscript) to demonstrate how this value might be applied to understanding the changing landscape of the humoral immunity to the escape variants in vivo.

Given its significance to the field it is important to share these findings and methodology with the research community for further validation in different groups and settings.

The paper could be strengthened by: i) The application of a retrospective analyses to understand the immunity generated in humans and monkeys following vaccination, and to further correlate these findings with protection outcomes.

ii) Inclusion of data from a panel of patient serum/plasma samples that had been previously ranked for breadth and potency by other methods to see how this approach compares.

iii) Clarifying how this approach is better than simply using the geometric mean titer (GMT). Figure 5 shows very clearly that the NP and GMT track very closely so more discussion of the added value of using the NP is needed.

iv) Introduction, first paragraph: Suggest replacing "complicated" with "requires standardization".

v) Last line of the introductory paragraph: Since Env panels vary a lot between labs, the inclusion of the something to the effect that "serum breadth and potency therefore depend strongly on the Env panels used that vary markedly between studies” would make that sentence more complete.

vi) Second paragraph of the Introduction: Minimise the repetition in discussing that tier 2 viruses are more difficult to neutralize.

vii) Subsection “Case studies”, last paragraph: Make it clear that this data is using monoclonal antibodies.

viii) Clarify what a concentration series is?

ix) Figure 1: It would be helpful to show the NP scores in blocks a and b as is done for block d. Is the scale between NP 1, 2 and 3 not linear? The spacing between the numbers on the x-axis suggests they are not.

x) Figure 6: The labelling is confusing. The same symbol is used to identify individual antibodies as well as sets of antibodies.

---

## [Author Response]

The paper could be strengthened by: i) The application of a retrospective analyses to understand the immunity generated in humans and monkeys following vaccination, and to further correlate these findings with protection outcomes.

We looked for suitable data sets among recent publications; there is little available to date in terms of Tier 2 neutralization responses with breadth in response to vaccination in human and monkeys. Thus, we have added an analysis of vaccinated rabbit sera from Rogier Sanders’ lab. Those neutralization data included 15 Envs that we could use to compute NP values. As detailed in the revised Results “This [ZM197M SOSIP.v5.2] immunogen induced Tier 2 responses in 4 of 5 test subjects, and yielded the most promising outcome among the SOSIP immunogens studied […] NP analysis agreed with the other neutralization metrics considered, and was able to resolve apparent ties between animals with similar neutralization responses to different Envs.” We look forward to being able to further correlate these findings with protection outcomes in future work.

ii) Inclusion of data from a panel of patient serum/plasma samples that had been previously ranked for breadth and potency by other methods to see how this approach compares.

We agree and used for this purpose a panel of serum neutralization data from a comparative study of breadth in progressors and long-term non-progressors (LTNP). As discussed in the revised text, that study compared 25 LTNP sera with 78 progressors, and noted conspicuously lower breadth and geometric mean ID50 neutralization titers in LTNPs than progressors. Using 10 of the 20 Envs published in that study, we computed NP values and found them highly correlated with potency and breadth. Results from comparing NP values between groups similarly indicated highly significant differences. The strong correlation between NP values and geometric mean ID50s indicates that the NP values provide a useful measure, consistent with the gold-standard neutralization assay used currently, but that to repeat the experiments used for this comparison using the NP method would require merely one-tenth the original number of neutralization reactions and sample material, given a five-point dilution series and testing 10 Envs, not 20.

iii) Clarifying how this approach is better than simply using the geometric mean titer (GMT). Figure 5 shows very clearly that the NP and GMT track very closely so more discussion of the added value of using the NP is needed.

Beyond being able to use greatly simplified and so more cost-effective experimental procedures to screen large-scale vaccine studies, as discussed above, we added this paragraph to the Discussion:

“Evaluating neutralization assay outcomes against the continuum of neutralization sensitivity among viral variants provides more context to interpret results, because it considers not only the proportion of Tier 2 Envs neutralized (breadth), but which Envs should most likely be neutralized. […] It helps also to compare sera where titers may be averaged over many outcomes below the limit of assay quantification, as was the case for sera from vaccinated rabbits.”

iv) Introduction, first paragraph: Suggest replacing "complicated" with "requires standardization".

Done.

v) Last line of the introductory paragraph: Since Env panels vary a lot between labs, the inclusion of the something to the effect that "serum breadth and potency therefore depend strongly on the Env panels used that vary markedly between studies” would make that sentence more complete.

Done.

vi) Second paragraph of the Introduction: Minimise the repetition in discussing that tier 2 viruses are more difficult to neutralize.

Done. (Deleted “which are more difficult to neutralize”)

vii) Subsection “Case studies”, last paragraph: Make it clear that this data is using monoclonal antibodies.

We added “monoclonal” to the final Results paragraph, the relevant Discussion paragraph, and the legend for Figure 6.

viii) Clarify what a concentration series is?

Replaced “antibody concentration series” with “antibody titration experiments”.

ix) Figure 1: It would be helpful to show the NP scores in blocks a and b as is done for block d. Is the scale between NP 1, 2 and 3 not linear? The spacing between the numbers on the x-axis suggests they are not.

We agree and have reformatted this notional figure to present the concept more clearly. The x-axis scale is based on rank, but purely speculative, so we have taken your advice.

x) Figure 6: The labelling is confusing. The same symbol is used to identify individual antibodies as well as sets of antibodies.

We have added symbols to this figure and explanatory text in the figure legend to identify which antibodies occur where.